# Mitochondria-related genes as prognostic signature of endometrial cancer and the effect of MACC1 on tumor cells

Xuefen Lin[1¤a¤b☯]Jianfeng Zheng[1,2¤a¤b☯], Yanhong Li[1,3¤a¤c], Linying Liu[1], Qinying Liu[2], Jie Lin[1], Yang Sun ID[1,2¤a¤b]*

1 Department of Gynecology, Clinical Oncology School of Fujian Medical University, Fujian Cancer Hospital, Fuzhou City, Fujian Province, China, 2 Fujian Provincial Key Laboratory of Tumor Biotherapy, Clinical Oncology School of Fujian Medical University, Fujian Cancer Hospital, Fuzhou City, Fujian Province, China, 3 Fujian University of Traditional Chinese Medicine, Fuzhou City, Fujian Province, China

☯ These authors contributed equally to this work.
¤aCurrent Address: 420 Forma Road, Jin'an District Fuzhou City, Fujian Province, China
¤bCurrent Address: 420 Forma Road, Jin'an District Fuzhou City, Fujian Province, China
¤cCurrent Address: No. 1 Qiuyang Road, Minhou Street, Fuzhou City, Fujian Province
* sunyang@fjzlhospital.com

## Abstract

Mitochondria are essential organelles involved in cell metabolism and are closely linked to various metabolic disorders. In this study, we aimed to develop a prognostic model for endometrial cancer (EC) patients based on mitochondria-related genes (MRGs), and to investigate the role of MACC1 in EC. As shown in the graphic summary, we retrieved gene expression and clinical data from open-access databases. To construct a predictive signature, we applied the Lasso Cox regression algorithm to MRGs. The predictive performance, immune features, and anti-tumor response of the mitochondrial signature were evaluated through multiple algorithms. Additionally, expression levels of key genes were validated using quantitative Real-Time PCR and Western Blot. A total of 2030 MRGs were retrieved, and 267 were found to be prognostically relevant. Eight MRGs—MACC1, CMPK2, NDUFAF6, DUSP18, TOMM40L, MT-TP, SAMM50, and MAIP1—were identified to construct a prognostic signature for EC. The MRG signature demonstrated significant associations with drug sensitivity, immune therapy, and immune cell infiltration. Based on comprehensive bioinformatic analysis, MACC1 was identified as the most promising MRG candidate in EC. Systematic experimental validation, including both in vitro and in vivo approaches, demonstrated that MACC1 down-regulation significantly suppressed EC progression, highlighting its potential as a therapeutic target.

**Data availability statement:** The RNA sequencing profiles are able to be gained from The Cancer Genome Atlas (TCGA) (https://portal.gdc.cancer.gov/) , the Genotype-Tissue Expression (GTEx) project (https://www.gtex-portal.org/home/) and MitoCarta 3.0 database (https://www.broadinstitute.org/mitocarta/mitocarta30-inventory-mammalian-mitochon-drial-proteins-and-pathways). Further inquiries can be directed to the corresponding author.

**Funding:** This work was supported by the High-level Talents Training Project of Fujian Cancer Hospital (Grant No. 2022YNG04 to Yang Sun) and the Clinical Research Center for Precision Treatment of Gynecological Malignancies of Fujian Province (Grant No. 2022Y2015 to Yang Sun).

**Competing interests:** The authors have declared that no competing interests exist.

## Introduction

Endometrial cancer (EC) is one of the three most common malignancies in the female reproductive system [1]. The incidence of EC has been rising steadily due to increased life expectancy and lifestyle changes, making it more common among younger individuals [2]. Obesity, metabolic syndrome, estrogen excess, and genetic susceptibility constitute pivotal risk factors for EC [1]. Metabolic disorders, such as diabetes, are not only associated with the incidence of EC but also with its adverse pathological characteristics [3]. The primary approach for treating EC involves surgically removing the uterus and bilateral salpingo-oophorectomy. Lymph node dissection may be used as an adjunct or targeted therapy based on clinicopathological and molecular characteristics. Comprehensive management of EC has notably improved the overall prognosis for patients, particularly those with endometrioid endometrial carcinoma (EEC). However, despite current treatment modalities, the survival outlook remains notably poor for patients with recurrent and metastatic EC [4]. The current lack of reliable biomarkers or predictive models hinders accurate prognostication of patients with EC [5]. Therefore, it is imperative to investigate a precise prognostic model in order to enhance the overall outcome of patients with EC.

The mitochondrion is an organelle characterized by a double-membrane structure that is ubiquitous in most cells, serving as the primary site for cellular energy production through aerobic respiration. Mitochondria take up substrates from the cytoplasm and use them for important cellular processes [6]. Mitochondria function as metabolic sensors in cells, exerting control over cancer cell apoptosis and initiating signaling cascades associated with cancer cell migration, invasion, metastasis, and resistance to treatment [7]. The communication and coordination between mitochondria and cells exert a significant impact on cellular metabolism and growth. The modulation of various regulatory pathways in mitochondria can exert an impact on tumor metabolic reprogramming and facilitate the proliferation of EC cells [8]. Furthermore, various aspects of mitochondrial biology, including mitochondrial biogenesis as well as fission and fusion dynamics, significantly contribute to tumorigenesis [9]. Otto Warburg postulated that mitochondrial respiratory deficiencies may underlie the phenomenon of aerobic glycolysis and its association with cancer [10]. Mitochondrial metabolism is a target for many cancer treatments [11]. However, growing evidence indicates that mitochondria play a complex and multifaceted role in cancer. For instance, mitochondria play pivotal roles in catabolism, anabolism, and signal transduction pathways during tumor progression [12]. As previously mentioned, the role of mitochondria in tumor growth has been established; however, the role of organelle-related genes in endometrial cancer is still unclear.

We conducted this study to determine if mitochondria-related genes (MRGs) can predict the prognosis of EC patients. Our results demonstrate that this prognostic signature accurately predicts patient outcomes and correlates with immunological features, tumor mutational burden (TMB), and chemotherapy sensitivity. Furthermore, we investigated the effect of MACC1 on EC cells.

## Materials and methods

### Data collection

We acquired the data from the Genotype-Tissue Expression (GTEx) project (https://www.gtexportal.org/home/, accessed February 10, 2024) [13] and The Cancer Genome Atlas (TCGA) (https://portal.gdc.cancer.gov/, accessed February 11, 2024) [14], including RNA sequences and clinical information for EC samples. We excluded primary EC samples with inadequate clinical data or follow-up information, ultimately including 545 primary EC samples and 113 healthy controls in our study. From these data, we extracted patient characteristics including age, pathological grade, pathological stage, and lymph node metastasis status for further analysis (S1 Table). Based on GENCODE's gene annotation data, the gene data was converted [15]. A total of 2030 MRGs (S2 Table) were downloaded from Mito-Carta 3.0 database (https://www.broadinstitute.org/mitocarta/mitocarta30-inventory-mammalian-mitochondrial-proteins-and-pathways, accessed February 12, 2024) [16]. The somatic mutations of mRNAs were generated using the maftools package, and their annotation was performed in Mutation Annotation Format (MAF) [17].

### Differential expression and prognosis evaluation for MRGs

We employed the limma package for differential expression analysis of MRGs in EC samples and normal tissues ($P < 0.05$ and $|\log FC| > 1$) [18]. The prognostic MRGs were identified through the utilization of Univariate Cox regression analysis, which involved assessing the levels of variable MRGs, survivor duration, and life state.

### Creation of the signature and cluster related to MRGs

The non-negative matrix factorization (NMF) clustering algorithm [19] was employed to classify EC patients based on differentially expressed MRGs that have prognostic significance. The MRGs used for risk score can be found in the supplementary information S3 Table. The MRG-signature presented herein was derived by integrating the expression levels of individual MRGs with prognostic coefficients obtained through LASSO regression analysis based on the formula [20]:

$$Risk\ score = \sum \beta gene \times Expgene$$

The $\beta_{gene}$ represented regression coefficient in the formula, while $Exp_{gene}$ signified the MRG's expression level. The risk score of each patient was determined, the median risk score was used as a cutoff to classify patients into high-risk and low-risk groups (high-risk if ≥ median, low-risk if < median). To validate the prognostic accuracy of the MRG-signature, we performed a random and equal allocation of total samples (Total Set) into Train and Test sets. We assessed survival outcomes for different risk groups using Kaplan-Meier curves. We evaluated the prediction model's efficacy using receiver operating characteristic (ROC) curves and both univariate and multivariate Cox regression analyses. We used the calibration curve to assess the accuracy of the visual nomogram [21]. Additionally, the effectiveness of the MRG-signature in distinguishing between patients with EC was assessed using principal component analysis (PCA) [22] and t-distributed stochastic neighbor embedding (t-SNE) [23].

### Enrichment pathways between the two subgroups

The clusterProfler package [24] was used to perform enrichment analysis for Kyoto Encyclopedia of Genes and Genomes (KEGG) and Gene Ontology (GO). The GSVA package [25] was used to perform Gene Set Variation Analysis (GSVA). The high-risk group underwent Gene Set Enrichment Analysis (GSEA) pathway enrichment analysis in comparison to the low-risk group [26].

 

## Assessment of immunological response and treatment

The IOBR approach was employed to quantify the quantity of immune-infiltrating cells [27]. Differences in immunotherapy between the two risk groups were assessed using submaps [28]. The oncoPredict software was used to assess the drug susceptibility of chemotherapeutic medications for patients with EC [29].

## Screening notable MRGs of EC with machine learning-dependent integrative methods

After validation of 12 machine learning techniques and 113 algorithm combinations, we extracted the most important MRGs from the MRG-related signatures, which enabled us to accurately and reliably validate MRGs between EC and normal samples [30]. Based on the in-depth analysis of its occurrence in 113 algorithm combinations, the MRG with the most frequency in the model was selected for in vivo and in vitro experiments.

## Cell culture

Meisen Chinese Tissue Culture Collections (Zhejiang, China) provided the commonly utilized EC cell lines HEC-1A, Ishikawa, RL95–2 and KLE, as well as human endometrial stromal cells (HESCs). Ishikawa cells were cultured in 90% RPMI 1640 medium plus 10% FBS, while HEC-1A cells were grown in 90% McCoy's 5A media plus 10% FBS (PAN-Biotech, Germany). RL95–2 and KLE cells were cultured in 90% DMEM medium plus 10% FBS. Cells were cultured at 37°C in a 5% $CO_2$ atmosphere.

## Quantitative real-time PCR

Total RNA was extracted from cells using RNAeasy™ Animal RNA Isolation Kit with Spin Column (Tiangen Biotech, Beijing, China). Reverse transcription was conducted following the instructions provided with FastKing gDNA Dispelling RT SuperMix. Quantitative real-time PCR (qRT-PCR) was performed with SuperReal PreMix Plus. Primer sequences are listed in S4 Table. The expression level was calculated by $2^{-\Delta\Delta Ct}$, the most commonly used method.

## Cell transfection

To reduce MACC1 expression, we employed GP-transfect-Mate (GenePharma, Shanghai, China) and small interfering RNA (siRNA) targeting MACC1 RNAs as well as negative control RNAs following the manufacturer's instructions. S5 Table provided siRNA sequences.

## Western blot

Total protein was extracted with RIPA buffer and its concentration was determined. Proteins were transferred to polyvinylidene fluoride membranes after being separated by sodium dodecyl sulphate polyacrylamide gel electrophoresis. Following an hour of blocking at room temperature, the membranes were incubated with primary antibodies against β-Actin (dilution 1:10000, Catalogue# EM21002, HUABIO, China) and MACC1 (dilution 1:2000, Catalogue# ER65531, HUABIO, China) for an entire night at 4°C. Secondary antibodies were added after washing and left for an hour at room temperature (dilution 1:10000, ZB-2306, ZSGB-BIO, China). To identify the protein bands, an improved chemiluminescent substrate was used. The ImageJ program was used to quantify the immunoblots.

## CCK-8 assays

Following transfection, the cells were cultured for a period of two to four days in 96-well plates containing 2000 cells per well and supplemented with 100μl of complete culture medium. The optical density (OD) at 450 nm was measured using an automated microplate reader (Molecular Devices, Suzhou, China) at 0, 24, 48, 72, and 96 hours with the Counting Kit-8 (APExBIO, Houston, TX, USA). Each group was tested in triplicate, with three wells per group.

## Scratch wound-healing assays

When cells were overfilled on the 6-well plate, that is, there is no space between the cells, gently scrape with a 200μl pipette tip and wash with PBS. Serum-free media was used to cultivate EC cells. Then, at time points 0 and 48 hours, three non-overlapping views in each hole were randomly taken. The image was processed using Image J, and the rate of wound healing was determined. Its calculation formula is: wound healing rate = $(A_{0h}-A_{48h})/A_{0h}\times100\%$, where $A_{0h}$ is the original area of the wound and $A_{48h}$ is the remaining area of the wound after healing.

## Transwell migration and invasion assays

The transwell assay for invasion and migration was conducted using polycarbonate-filtered chambers with 8 μm pore sizes. In the upper compartment, 300μL of serum-free medium was utilized to cultivate the $2\times10^5$ EC cells, whereas in the lower chamber, 500μL of cell medium containing 30% FBS was added. For the invasion experiment, the chamber is coated with a layer of Matrigel. After 48 hours of development at an uninterrupted temperature, any cells that did not fit through the pores were carefully removed using a cotton swab. The chamber was stained with 0.1% crystal violet. This was applied for five minutes following a ten-minute fixation with 4% paraformaldehyde. Using a microscope, the quantity of cells on the lower surface was then counted quantitatively.

## Colony formation assays

A total of 500 cells were seeded in each well of 6-well culture plates and incubated for a period of 14 days. Upon the appearance of discernible clones, the supernatant was carefully withdrawn. Subsequently, two washes with PBS were performed, followed by addition of 1 mL of a 4% paraformaldehyde fixative and incubation for 15–20 minutes to achieve fixation. After discarding the supernatant and washing it twice with PBS, add 1 mL crystal violet dyeing solution for 15 minutes. After gently rinsing the culture plates with tap water, allowing them to air dry, and subsequently capturing images, the visible clones were counted.

## Apoptosis assays

As directed by the manufacturer, MedChemExpress, NJ, USA, apoptosis assays were carried out. When the trans-fected cells in the culture vial grew to more than 80%, they were digested with EDTA-free pancreatic enzyme and then re-suspended with 195μL Binding Buffer. The cells were stained with 5μL propyl iodide (PI) and 10μL Annexin V-FITC. After incubation in the dark at room temperature for 15 minutes, the apoptosis rate was quantified by flow cytometry (BD Biosciences, New York, USA).

## EdU cell proliferation assays

In 24-well plates, $2\times10^5$ MACC1 knock-down EC cells and their corresponding control cells were seeded, and the cells were cultured for a complete day. Using the EdU analysis kit (APExBIO, Houston, TX, USA), cell proliferation was exam-ined and assessed, and the particular process was followed in compliance with the manufacturer's recommendations.

## Animal experiments

The animal protocols were approved by the Institutional Animal Ethical Committee of Fujian Cancer Hospital, which specifically reviewed and authorized the experimental endpoints. Female nude mice (8–10 weeks old, n = 30 total) from Fujian Medical University's experimental animal center were housed under controlled conditions (22 ± 2°C, 50 ± 10% humidity, ad libitum food/water). Mice received subcutaneous injections of $1\times10^6$ MACC1-siRNA-NC or MACC1-siRNA cells (100 μL PBS) into the right flank. Tumor volume (V=(length×width²)/2) was measured weekly for 4 weeks. Humane endpoints included tumor burden exceeding 15% of body weight, ulceration, or severe distress; mice meeting these

criteria or completing the study were euthanized via $CO_2$ inhalation (no unexpected deaths occurred). Health and behavior were monitored daily. Analgesics were not required as the model involved minimal discomfort. All tumors were excised, weighed, and stored in liquid nitrogen post-euthanasia.

## Statistical analysis

Statistical analysis was performed using R (v 4.0.2) or GraphPad Prism (v 9.0). Results, presented as Mean ± SD, were based on triplicate assays. Data normality was assessed using normality tests available in GraphPad Prism. Using log-rank testing and Kaplan-Meier curves, survival results between subgroups were compared. For the relationships between the variables, the Spearman coefficient for correlation was employed. The 2-tailed Student's t-test was used to compare data between two groups, and a one-way ANOVA was employed for groups in excess of two. Each section contains a deep analysis of that particular topic. At $P < 0.05$, statistical significance was established.

## Results

### Expression and functional analysis of MRGs

A total of 1,009 differentially expressed MRGs were identified between EC and healthy tissues (Fig 1A), including 805 upregulated and 204 down-regulated MRGs (|log2FC| > 1, adjusted p-value < 0.05). The functionalities of the 1009 MRGs were examined using GO and KEGG pathway enrichment analysis (Fig 1B-C). Evidently, there was an enrichment observed in several terms associated with Biological Process (BP), Cellular Component (CC), and Molecular Function (MF) among the 1009 differential MRGs of EC. These terms encompassed energy derivation through organic compound oxidation, aerobic respiration, apoptotic mitochondrial alterations, and other related phenomena (Fig 1B). Additionally, the 1009 MRGs that were found to be differentially expressed were highly enriched in a number of metabolic pathways, as demonstrated by KEGG analysis. These pathways included the metabolism of carbon, the citrate cycles cycle (TCA cycle), fatty acids, 2-oxocarboxylic acid, lipoic acid, propanoate, and pyruvate, among various others (Fig 1C).

### Molecular cluster based on MRGs

The EC molecular subtypes were determined using NMF method. EC Patients were classified into MRG-related Cluster 1 (C1) and Cluster 2 (C2), as the optimal number of clusters, determined through cophenetic, dispersion, and silhouette analysis, was found to be two (S1 Fig). Fig 1D demonstrated that patients with EC in C1 exhibited a more favorable prognosis, whereas those in C2 display a poorer prognosis. To elucidate the biological functionality of MRG-related clusters, we conducted an assessment of differentially expressed genes (DEGs) between C1 and C2. Based on the analysis of GO terms (Fig 1E) and KEGG pathways (Fig 1F), we identified a significant association between these DEGs and crucial biological processes, including nuclear division, DNA replication, Cell cycle, RNA localization, DNA−templated DNA replication, Fatty acid metabolism, Fatty acid degradation, N−Glycan biosynthesis, among others.

### Construction and validation of the risk model

A prognostic MRG-related signature was created via LASSO Cox analysis applied to the 267 prognostic MRGs. In Fig 2A-B, the λ selection diagram was displayed. The risk model was developed by selecting a total of eight MRGs (MACC1, CMPK2, NDUFAF6, DUSP18, TOMM40L, MT-TP, SAMM50, and MAIP1) for analysis (Fig 2C). Survival analysis identified six MRGs (MACC1, CMPK2, NDUFAF6, TOMM40L, MT-TP, and MAIP1) as risk factors (HR > 1) and two MRGs (DUSP18 and SAMM50) as protective factors (HR < 1) (Fig 2C). Based on their mean risk scores, EC patients in the Train, Test, and Total sets were separated into low-risk and high-risk categories. Kaplan-Meier curves were used to evaluate the correlation between risk scores and survival outcomes. Risk score distributions for the two groups are shown in S2 Fig. Whether in the training, test, or total sets, the overall survival (OS) of EC patients in the high-risk

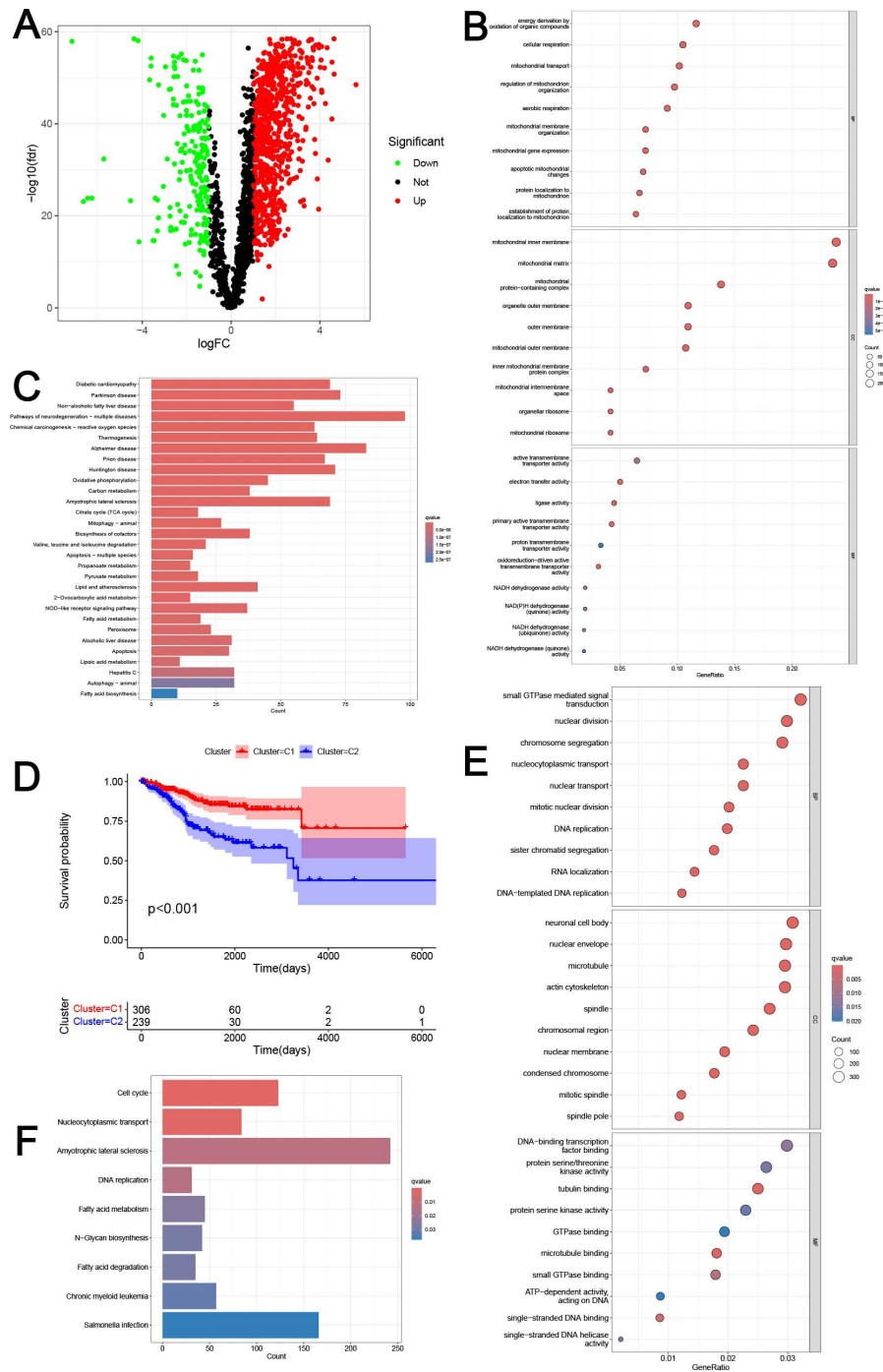

**Fig 1. Expression and functional analysis of MRGs. (A)** Volcano map of differentially expressed MRGs in EC. The red dots represented highly expressed genes, and the green dots represented low-expressed genes. **(B)** The GO functional enrichment analysis of the differential MRGs. The size of the dots indicates the number of genes attributed to the corresponding category. The color of the dots represented the q value. **(C)** KEGG pathway analysis of the differentially expressed MRGs. The color of the bars represented the q value. **(D)** The survival curves of the molecular subtypes. The red curve represented Cluster I and the blue curve represented Cluster **II. (E)** The GO functional enrichment analysis of the differentially expressed genes between Cluster I and Cluster **II.** The size of the dots indicates the number of genes attributed to the corresponding category. The color of the dots represented the q value. **(F)** KEGG pathway analysis of the differentially expressed genes between Cluster I and Cluster **II.** The color of the bars represented the q value.

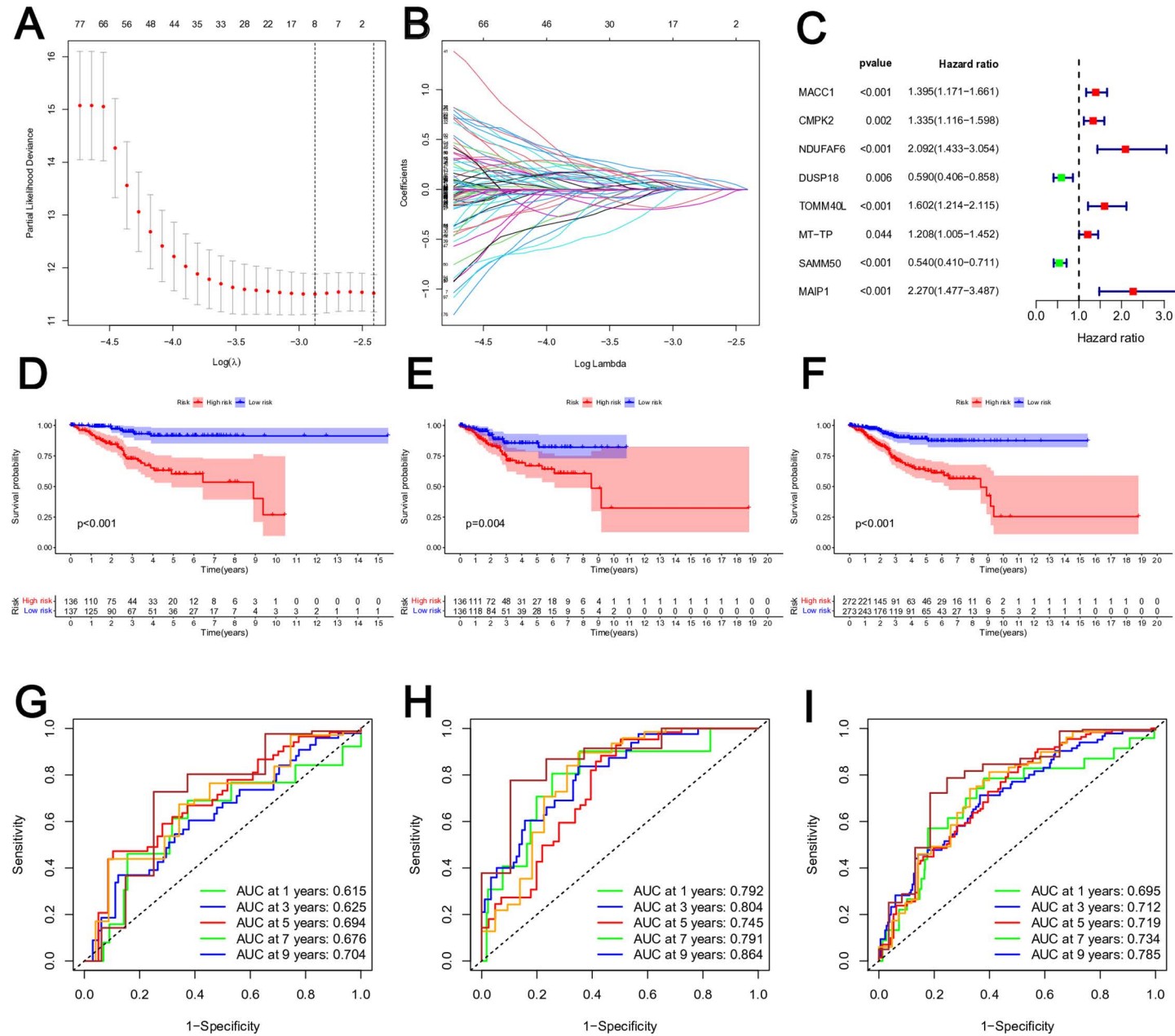

**Fig 2. Construction and validation of the risk model based on MRGs. (A)** Cross-validation for tuning the parameter selection in the LASSO regression. **(B)** LASSO regression of the 8 prognostic MRGs. **(C)** The 8 prognostic MRGs extracted by Univariate Cox regression analysis were shown in the forest map. **(D-F)** The K-M survival curves of train set**(D)**, test set**(E)** and total set **(F)** based on the MRG-related risk model. The red curve represented the high-risk group, and the blue curve represented the low-risk group. **(G-I)** Time-dependent ROC curve analysis of train set**(G)**, test set**(H)** and total set **(I)**.

group was substantially lower than that in the low-risk group, as Fig 2D–F illustrated. Consistent progression-free survival (PFS) data further corroborated the risk model's survival prediction accuracy (S3 Fig). The MRG-related signature's resilience in forecasting survival for one, three, five, seven, and nine years was shown by the time-dependent ROC curve in the train set, test set, and total set (Fig 2G–I).

## Comparison of clinical features and MRG-related signature

Six MRGs (MACC1, CMPK2, NDUFAF6, TOMM40L, MT-TP, and MAIP1) showed higher expression levels in high-risk EC patients, whereas two MRGs (DUSP18 and SAMM50) had lower expression levels in low-risk EC patients (Fig 3A, S4 Fig). High-risk EC patients were generally older, had more advanced stages and grades, and a higher incidence of lymph node metastasis (LNM) positivity (Fig 3B). Moreover, older patients tended to have higher risk scores (Fig 3C), and those with higher grades exhibited increased risk levels (Fig 3D). Furthermore, C1 was associated with elevated risk scores (Fig 3E), as well as higher Stage indicating greater risk (Fig 3F). Additionally, patients with LNM displayed heightened risk

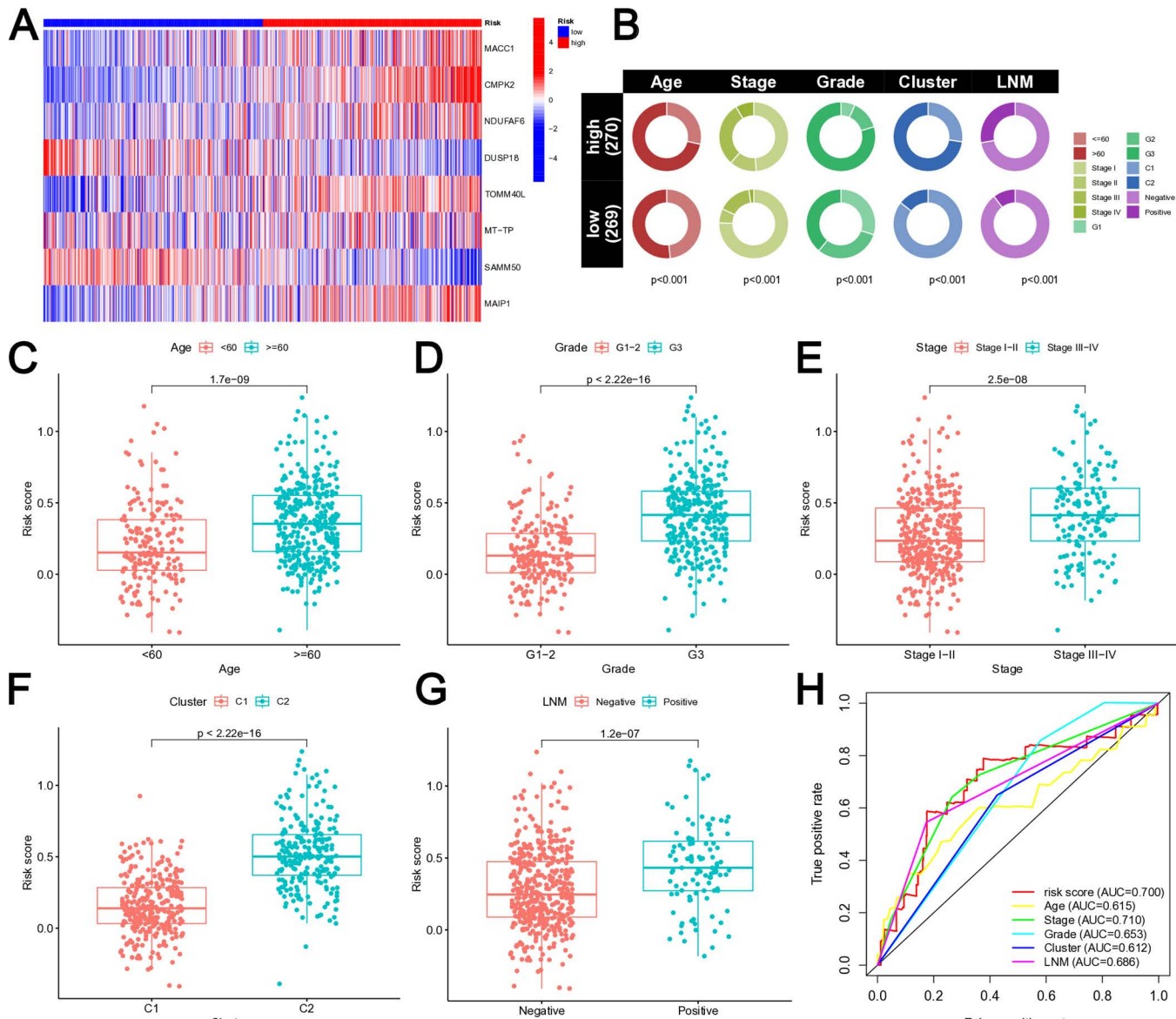

**Fig 3. Comparison of clinical features and MRG-related signature. (A)** Heatmap of the expression levels of the 8 MRGs contained in the MRG-related signature. **(B)** The pie chart showing the proportion of patients in the two risk groups for each clinical feature. **(C-F)** Differences in risk scores among the clinical features, including Age **(C)**, Grade **(D)**, Cluster **(E)**, Stage **(F)**, and LNM **(G)**. **(H)** The AUC of risk score and clinical characteristics.

scores (Fig 3G). The ROC curve demonstrated that the area under the curve (AUC) for the MRG-related signature was 0.7 (Fig 3H), surpassing the AUCs of Age, Grade, Cluster, and LNM. This observation suggested that the MRG-related signature exhibited a higher level of confidence and possesses stronger predictive potential.

Additionally, the expression levels of the EC prognostic MRGs employed in model construction exhibited a substantial capacity to discriminate EC patients, as indicated by PCA (Fig 4A) and t-SNE (Fig 4B) analyses. Furthermore, both Univariate and Multivariate Cox regression analysis demonstrated that the risk model was an independent prognostic factor for individuals with EC (Fig 4C-D). The nomogram diagram (Fig 4E) facilitated a visual representation of the outcomes. To

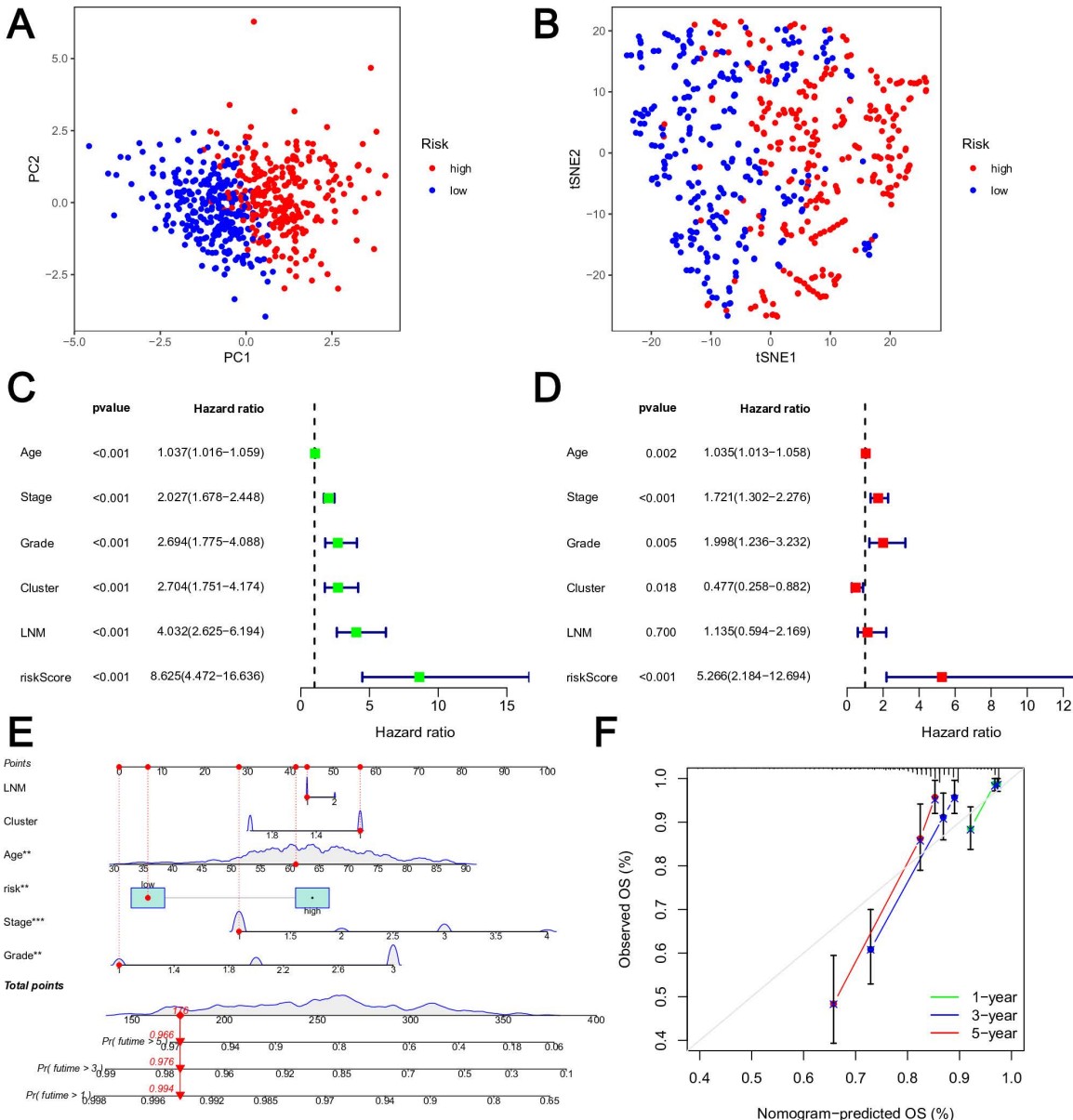

**Fig 4. Clinical value of risk score by independent prognostic analysis. (A)** PCA analysis of the two risk groups. **(B)** t-SNE analysis of the two risk groups. **(C)** The Univariate analysis of risk model and clinical features. **(D)** The Multivariate analysis of risk model and clinical features. **(E)** The Nomogram model based on risk model and clinical features. **(F)** The calibration curve of the risk model.

verify the model's accuracy, a calibration curve was constructed (Fig 4F). A closer alignment between the curve and either the 45-degree line or the gray lines on the graph indicated a stronger fitting effect.

To comprehensively validate the prognostic robustness of our MRG-related signature, we conducted stratified survival analyses across key clinicopathological features, including age (<60 vs. ≥60 years, S5 Fig A-B), tumor grade (G1-2 vs. G3, S5 Fig C-D), FIGO stage (I-II vs. III-IV, S5 Fig E-F), MRG-related clusters (S5 Fig G-H), and lymph node status (positive vs. negative, S5 Fig I-J). Notably, the signature maintained its prognostic significance within each subgroup, consistently demonstrating significantly poorer overall survival in high-risk patients compared to their low-risk counterparts (all $P < 0.05$).

## Functional pathways and mutation evaluation

Functional pathway enrichment analysis based on the GSVA method allowed us to find 31 pathways where there were substantial variations between the two risk groups (Fig 5A). The low-risk group demonstrated significant enrichment in multiple metabolism-related pathways, including PRIMARY BILE ACID BIOSYNTHESIS, ALPHA LINOLENIC ACID METABOLISM, ETHER LIPID METABOLISM, LINOLEIC ACID METABOLISM, FRUCTOSE AND MANNOSE METABOLISM, FATTY ACID METABOLISM, and GLYCOSAMINOGLYCAN DEGRADATION TYROSINE METABOLISM (Fig 5A). The low-risk group was enriched in 80 pathways, while the high-risk group was considerably enriched in 105 pathways, according to the GSEA enrichment analysis. After prioritizing the pathways enriched in risk groups and selecting the top five pathways for visualization, we found a strong correlation between different metabolic activities and low risk-scores, including FATTY ACID METABOLISM, GLYOXYLATE AND DICARBOXYLATE METABOLISM and TYROSINE METABOLISM (Fig 5B). Besides, the high risk-scores were associated with ASCORBATE AND ALDARATE METABOLISM, PORPHYRIN AND CHLOROPHYLL METABOLISM, PENTOSE AND GLUCURONATE INTERCONVERSIONS and STARCH AND SUCROSE METABOLISM (Fig 5C). We analyzed DEGs between high-risk and low-risk groups to explore the biological functions of the MRG-related signature. GO analysis showed that these DEGs were significantly enriched in several key biological processes (Fig 5D).

Tumor mutation burden (TMB) was larger in the low-risk group than in the high-risk group, according to our analysis of the TCGA-UCEC cohort's mutational data. (Fig 5E). The TMB score quartile was utilized to stratify EC patients into low-TMB and high-TMB cohorts. Our findings revealed that patients diagnosed with EC exhibited susceptibility to both elevated risk scores and diminished TMB levels (Fig 5F). The waterfall diagram (Fig 5G-H) depicted the top 15 mutant genes of the two MRG-related risk groups, implying a discernible molecular distinction between these two cohorts. The low-risk group exhibited the highest TP53 mutation rate (Fig 5G), whereas the high-risk group displayed the highest PTEN mutation rate (Fig 5H).

## Assessment of immune activity and immunotherapy

To systematically characterize the immune landscape associated with our risk signature, we employed seven complementary computational deconvolution methods (CIBERSORT, EPIC, ESTIMATE, MCPcounter, quanTIseq, TIMER, and xCell) to profile the tumor immune microenvironment (S6 Fig A). This multi-algorithmic approach enabled robust and comprehensive evaluation of immune cell infiltration patterns. Our analysis revealed striking differences in immune cell composition between high- and low-risk groups. Notably, we observed significant enrichment of multiple immune cell populations in the high-risk group, particularly in adaptive immune components. Specifically, both CD4+and CD8+T cell populations showed consistent elevation across multiple algorithms (MCPcounter_T_cells, MCPcounter_CD8_T_cells), suggesting enhanced T cell infiltration in high-risk tumors. Intriguingly, we observed substantial differences in stromal components, particularly in endothelial cells and cancer-associated fibroblasts (EPIC_CAFs, EPIC_Endothelial, MCPcounter_Endothelial_cells, xCellmv_Endothelial_cells). This finding suggests that the risk signature not only reflects immune cell dynamics but also encompasses broader changes in the tumor microenvironment. The ESTIMATE scores further corroborated these

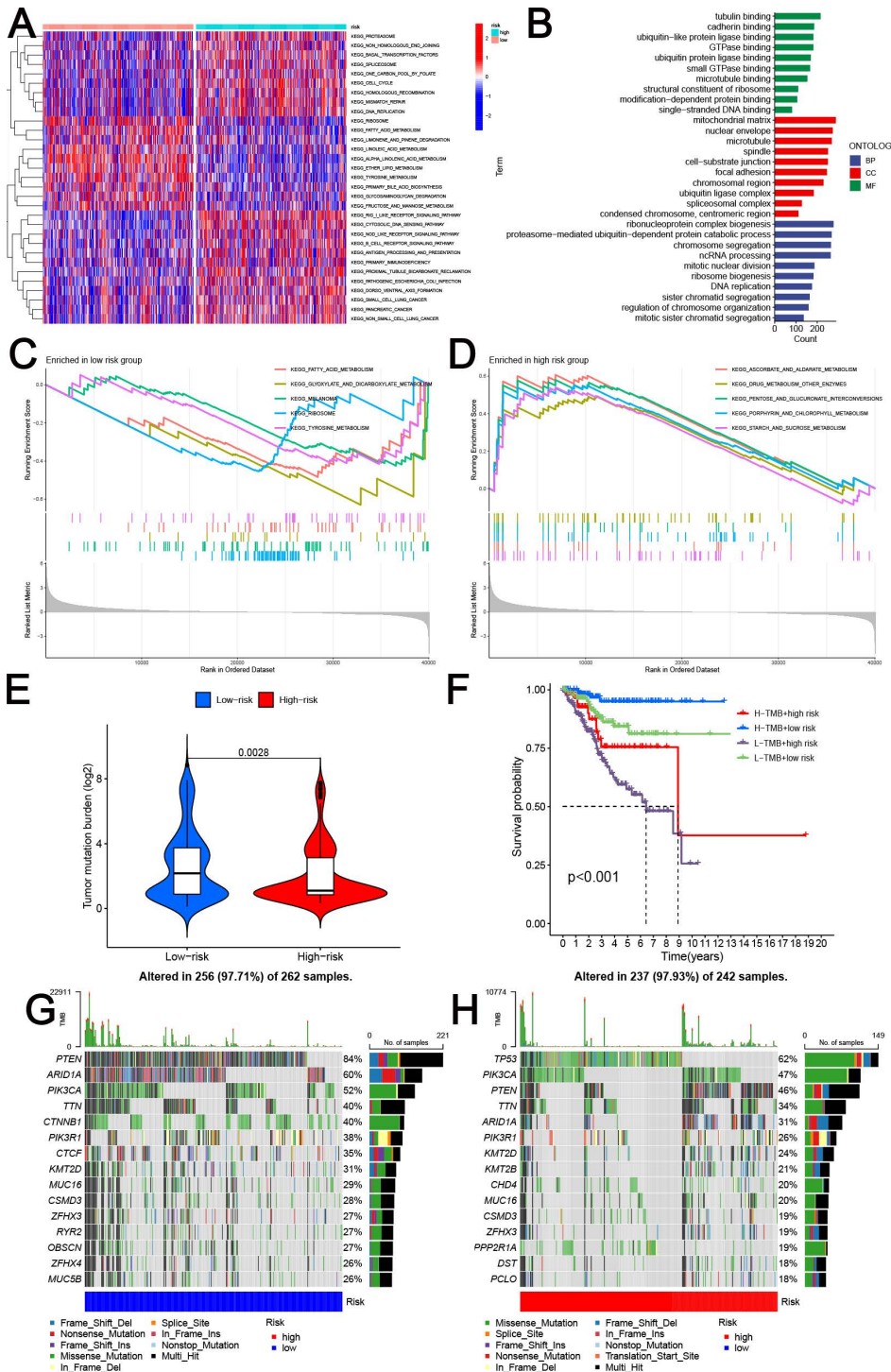

**Fig 5. Functional pathways and evaluation of mutation between the two MRG-related risk groups.** **(A)** The GSVA analysis of two risk subgroups. **(B)** GO analysis between high-risk and low-risk groups. **(C)** The top five significant enrichment pathways in the low-risk group by GSEA enrichment analysis. **(D)** The top five significant enrichment pathways in the high-risk group by GSEA enrichment analysis. **(E)** The level of TMB between high-risk and low-risk groups. **(F)** Survival analysis of distinct groups stratified by both TMB and signature. **(G-H)** The waterfall plot of somatic mutation features established with low **(G)** and high **(H)** risk scores.

findings, indicating significant differences in both immune and stromal components between risk groups. Of particular interest was the enhanced NK cell signature (MCPcounter_NK_cells) in the high-risk group, suggesting increased natural cytotoxicity potential. This observation, combined with the elevated T cell signatures, indicates a more active but potentially dysregulated immune response in high-risk tumors. Notably, these findings suggest that high-risk patients might be particularly responsive to immunotherapy approaches, specifically PD1-R therapy, as supported by our additional analysis (S6B Fig).

### Drug susceptibility analysis

We explored the drug susceptibility between the two risk groups using data from the GDSC database. In addition to Cisplatin (Fig 6A), the high-risk group exhibited significantly elevated IC50 levels for commonly employed chemotherapy drugs compared to the low-risk group, including Crizotinib (Fig 6B), Cytarabine (Fig 6C), Docetaxel (Fig 6D), Paclitaxel (Fig 6E), Tamoxifen (Fig 6F), Sorafenib (Fig 6G) and Vinorelbine (Fig 6H). The results of this study indicate a negative correlation between high-risk score and drug susceptibility in EC. The relationship between the expression levels of the eight genes linked to the MRG-related signature and drug susceptibility was evaluated by computing the Spearman's correlation coefficients (Fig 6I). According to our findings, the genes in the model were associated with drug susceptibility. A positive correlation was observed between the levels of MACC1 and CMPK2 and the drug resistance scores calculated by oncoPredict for several medications, including vinorelbine, cytarabine, docetaxel, tamoxifen, sorafenib, and crizotinib (Fig 6I). This indicates that higher expression levels of MACC1 and CMPK2 may be associated with decreased drug sensitivity to these drugs. Conversely, an inverse relationship was identified between the levels of SAMM50 and NDUFAF6 and the drug resistance scores for certain medications, such as Paclitaxel and Sorafenib (Fig 6I). This suggests that higher expression levels of SAMM50 and NDUFAF6 may be linked to increased drug sensitivity to these drugs.

### Knockdown of MACC1 can inhibit tumor growth

The eight MRGs included in the MRG-related signature were further narrowed down to the most pertinent MRG for in vitro and in vivo experimental validation using a total of 113 combinations of machine learning techniques. The results demonstrated that all models exhibited high AUC values in both the Train, Test, and All sets (S7 Fig). Furthermore, it was observed that the 113 algorithm combinations exhibited the highest frequency of SAMM50, NDUFAF6, MACC1, and DUSP18 occurrences (S8 Fig A). Based on the TCGA database, we identified a significant upregulation in the expression levels of SAMM50, MACC1, DUSP18, and NDUFAF6 ($P < 0.0001$, S8 Figs B-E). Notably, among these four previously described MRGs, MACC1 exhibited the highest logFC value (logFC = 1.68) with statistical significance ($P < 0.0001$, S8 Fig C). Consequently, we selected MACC1 as the target for subsequent experimental validation. The results of q-RT PCR ($P < 0.001$) and western blot assays ($P < 0.01$) revealed a significantly elevated expression level of MACC1 in EC cells (Figs 7A-B, Fig S9). To investigate the effect of MACC1 on tumor, we transfected EC cells with MACC1-siRNA. MACC1-siRNA-2 (si2) showed better efficacy than MACC1-siRNA-1(si1) in the evaluation of knockdown efficiency (Figs 7C-D, Fig S10). The results of the CCK-8 assays ($P < 0.001$, Fig 7E), colony formation assays ($P < 0.05$, Fig 7F), and EdU assays ($P < 0.05$, Fig 7H) revealed a significant reduction in EC cells viability and proliferation associated with down-regulation of MACC1 expression. After two weeks of siRNA transfection, we re-validated the expression level of MACC1. The western blot results showed that siRNA maintained a good knockdown efficiency ($P < 0.05$, Fig 7G, Fig S11). The results of the scratch wound-healing assays ($P < 0.05$, Fig 8A) and the transwell assays ($P < 0.05$, Fig 8B) demonstrated a significant reduction in migratory and invasive capacity upon MACC1 knockdown. Furthermore, the apoptosis experiments revealed a significant increase in EC cells apoptosis ($P < 0.01$, Fig 8C) subsequent to MACC1 knockdown. During tumor growth, downregulation of MACC1 significantly inhibited tumor volume, resulting in reduced tumor weight and size, indicating that MACC1 down-regulation suppresses tumor growth ($P < 0.01$, Fig 8D). Consequently, MACC1 functions as an oncogenic driver that facilitates EC development.

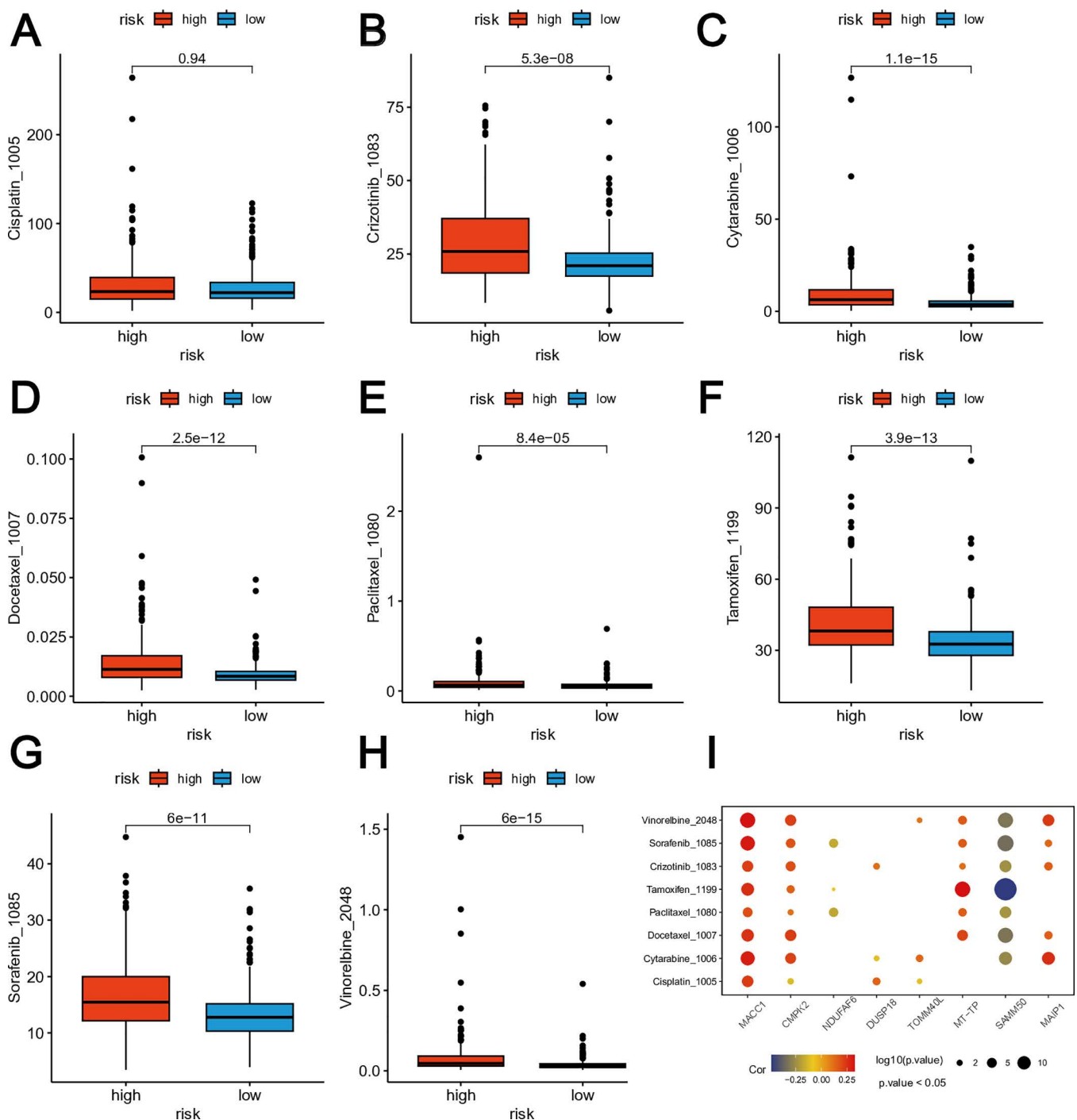

**Fig 6. The differences in the chemotherapy response of common chemotherapy drugs between the high- and low-risk groups. (A-F)** Relationships between risk scores and IC50 level of Cisplatin **(A)**, Crizotinib **(B)**, Cytarabine**(C)**, Docetaxel**(D)**, Paclitaxel**(E)**, Tamoxifen**(F)**, Sorafenib**(G)** and Vinorelbine **(H)**. **(I)** The Spearman's correlation coefficients between drug susceptibility and expression levels of the 8 genes in the MRG-related risk model.

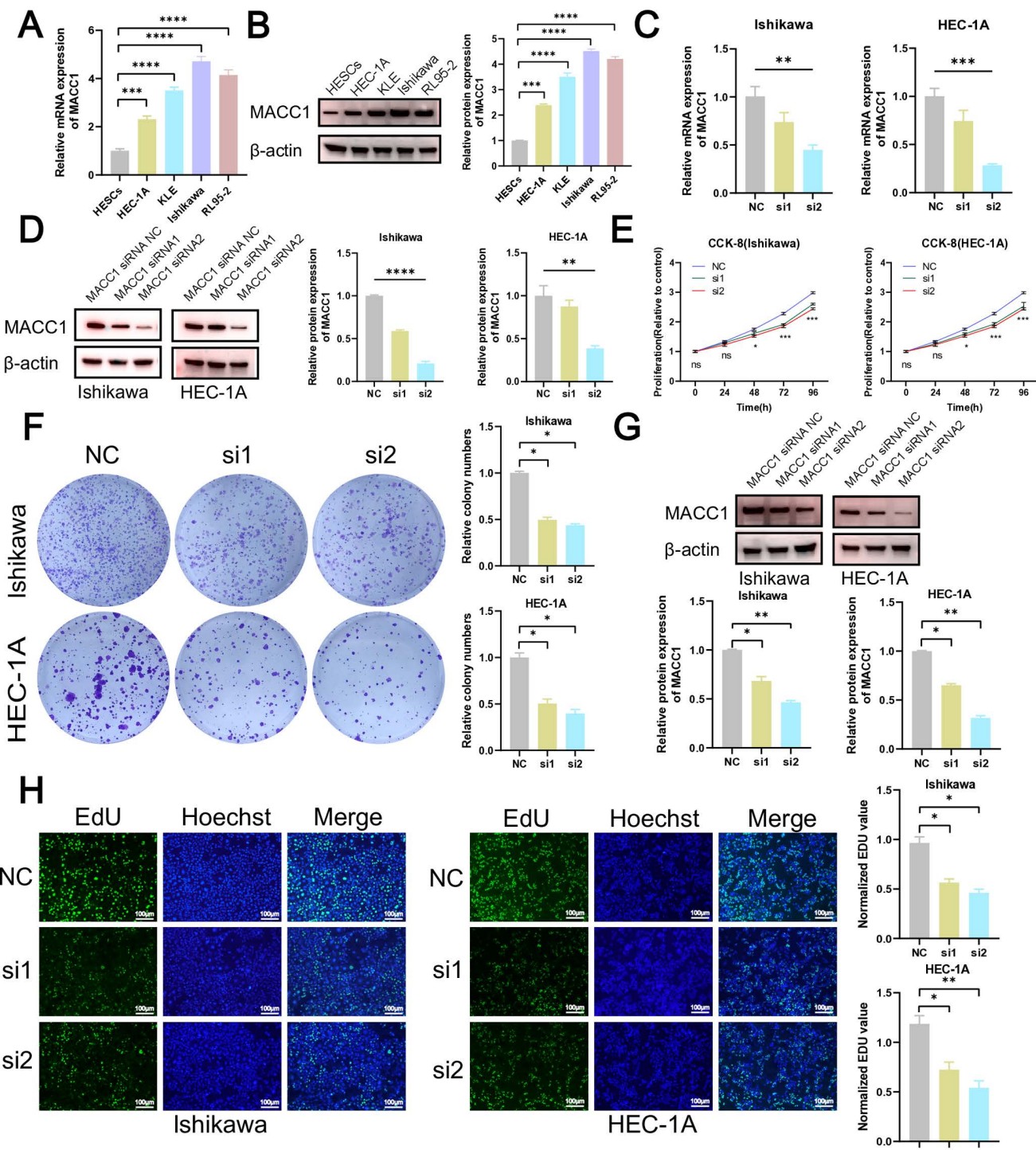

**Fig 7. Knockdown of suppresses the proliferation of EC cells. (A-B)** The expression of MACC1 in HESCs and EC cells. **(C-D)** The knockdown efficiency of MACC1 in EC cells. **(E)** CCK-8 assays of NC and the si2 groups to detect cell viability. The data marked with ns or asterisks were presented as mean±SD (n=3) and subjected to ANOVA analysis. ns: not significant, **$P < 0.01$, and ***$P < 0.001$ compared to the NC group at the respective time points (0h, 24h, 48h, 72h and 96h). **(F)** Colony formation assay. **(G)** The knockdown efficiency of MACC1 in EC cells after two weeks of siRNA transfection. **(H)** EdU staining were employed to assess cell proliferation. ns: not significant, *$P < 0.05$, **$P < 0.01$, ***$P < 0.001$, ****$P < 0.0001$.

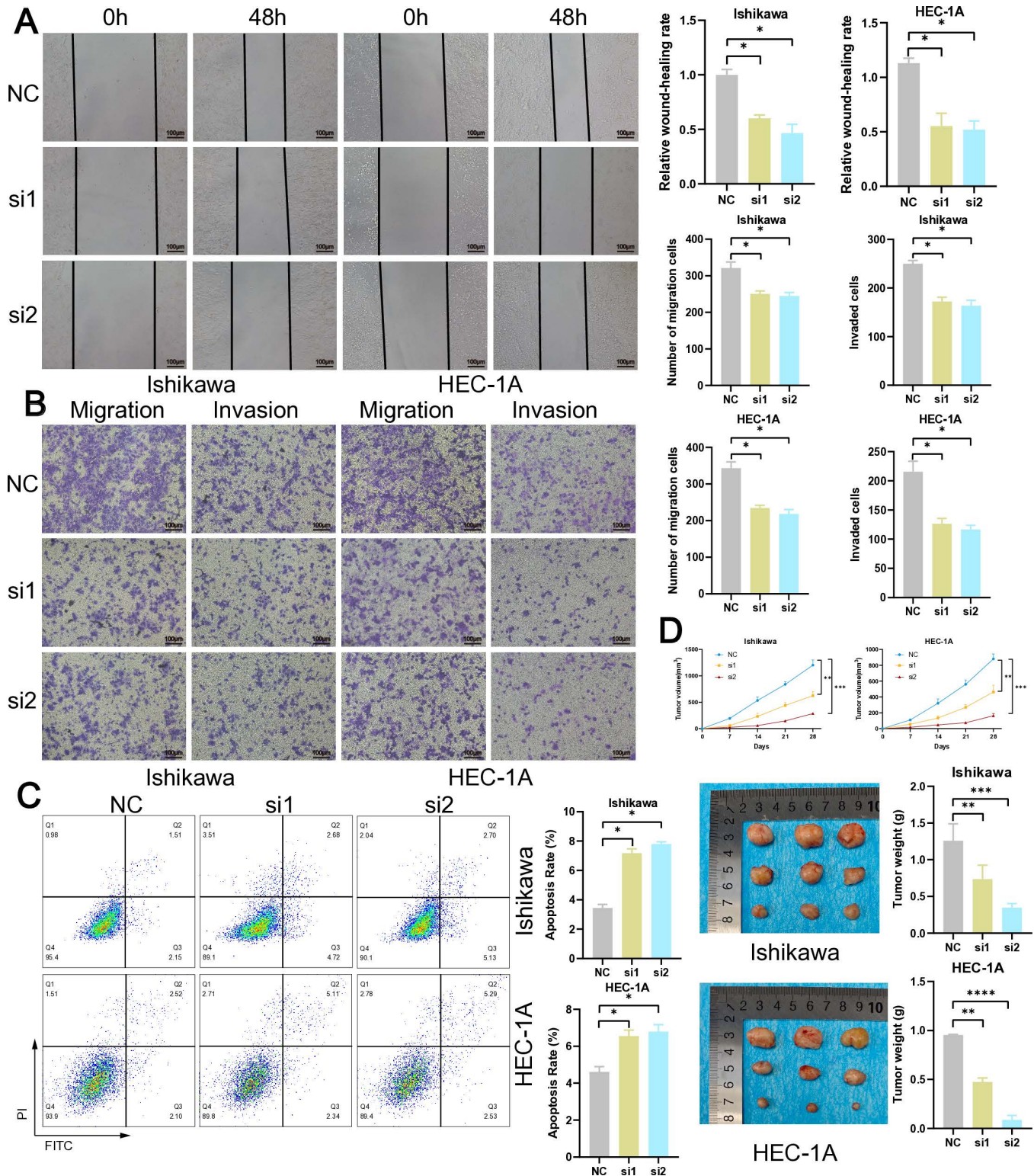

**Fig 8. Knockdown of suppresses the migration, invasion, and promotes the apoptosis of EC cells. (A)** Wound healing assay. **(B)** Cell migration and invasion measured through transwell assay. **(C)** Flow cytometry detected cell apoptosis. **(D)** Tumor volume, tumor weight and representative tumor images were shown to assess tumor growth. ns: not significant, *P<0.05, **P<0.01, ***P<0.001, ****P<0.0001.

## Discussion

Mitochondria are key intracellular organelles involved in energy production, cellular metabolism, and signal transduction. For instance, the generation of reactive oxygen species (ROS) within mitochondria can activate signal transduction pathways as a byproduct of electron transport chain (ETC) activity [31]. In normal cellular physiology, precise regulation of mitochondrial ROS (mROS) levels governs their involvement in diverse cellular processes, encompassing differentiation, autophagy, and metabolic adaptation [32,33]. Overproduction of ROS causes cellular death in cancer cells and promotes oncogenesis by causing genomic instability, changing gene expression, and initiating signalling cascades [34]. In ATP metabolism, mitochondria utilize pyruvate derived from glycolysis to generate ATP through oxidative phosphorylation. Additionally, in lipid metabolism, mitochondria possess the capability to synthesize phospholipids such as phosphatidy-lethanolamine (PE) and cardiolipin (CL), both of which are indispensable for maintaining mitochondrial respiratory function [35,36]. Mitochondrial DNA (mtDNA) mutations are frequently observed in neoplastic cells, and a diverse array of metabolites, particularly those within the mitochondria of cancer cells, govern tumor metabolism and progression [37]. It has been well studied how mitochondria function in tumors; however, there is a paucity of research on mitochondrial involvement in EC. Currently, no mitochondria-related prognostic markers exist to guide treatment decisions, improve prognosis prediction, prolong survival time, or improve the standard of living for EC patients.

EC metabolism has emerged as a crucial area of research in recent years, with studies revealing altered metabolic pathways that contribute to its development and progression. For instance, metabolic reprogramming, such as increased aerobic glycolysis and fatty acid synthesis, has been implicated in EC [10,38–40]. Eskander et al. discussed the integration of immunotherapy into first-line treatment for advanced and metastatic endometrial cancer, highlighting the evolving therapeutic strategies that potentially target metabolic pathways [41]. The significance of research in this area cannot be overstated, given the rising incidence and mortality of endometrial cancer, coupled with the limited success of current treatment options. By investigating the specific mechanisms through which metabolic pathways contribute to tumorigenesis and progression, researchers can identify potential therapeutic targets and develop innovative treatment strategies. This could pave the way for more effective and personalized treatments for endometrial cancer, ultimately improving patient outcomes.

In our study, we analyzed the expression profiles of 2030 MRGs in EC using data from the MitoCarta 3.0 database. With the use of MRGs with considerable prognostic value and differential expression, we were able to create useful prognostic characteristics. The functional pathway, TMB, somatic mutation features, immunological status, and chemotherapeutic drug sensitivity were compared between the two groups based on the MRG-signature. To enhance the prognosis of patients with EC and potentially identify novel therapeutic targets, our study proposes utilizing MRGs as the prognostic markers for the first time in order to predict the outcome of EC. To confirm the accuracy of the model, we also studied the genes in vivo and in vitro.

The gene Metastasis-Associated in Colon Cancer 1 (MACC1), discovered in 2009, exhibits a close association with the occurrence and metastasis of colon cancer. Stein et al. demonstrated a significant correlation between MACC1 and the incidence, invasiveness, and metastasis of various tumors [42]. MACC1 also contributes to the promotion of mesenchymal epithelial transition [43], and cancer immunotherapy[44]. In addition to inducing tumor cell migration, MACC1 is also closely related to cytoskeleton and adhesion system [44]. Our study demonstrates that down-regulation of MACC1 significantly curtails the migration and invasion of endometrial cancer cells. This inhibitory effect may be associated with MACC1's well-documented role in facilitating migration and metastasis through modulation of epithelial-mesenchymal transition (EMT) and cytoskeletal dynamics [45]. EMT is characterized by reduced E-cadherin expression and increased mesenchymal markers, and as a pivotal mediator of the transition from epithelial to mesenchymal cell states [46], the observed decrease in cellular motility following MACC1 knockdown supports its role in regulating EMT. MACC1 can influence the remodeling of the actin cytoskeleton [47], a regulation crucial for cellular morphology and migration. Our findings reveal that knockdown of MACC1 results in a decline in the invasive capacity of endometrial cancer cells, potentially

linked to MACC1's function in cytoskeletal organization. Furthermore, the reduction in tumor cell migration and invasion observed in our experiments aligns with MACC1's established role as a metastasis-promoting factor [46], providing a mechanistic basis for its prognostic value in endometrial cancer.

The involvement of MACC1 in the HGF/c-Met signaling pathway is a critical mechanism driving its oncogenic effects. MACC1 facilitates the EMT, cell invasion, and metastasis by regulating the HGF/c-Met signaling pathway [48]. This pathway, which promotes tumor cell survival and migration, is frequently upregulated in EC and other malignancies. Targeting the HGF/c-Met axis, along with MACC1, offers a promising strategy for inhibiting tumor progression. Notably, MEK inhibitors, such as trametinib, have shown efficacy in suppressing the HGF/c-Met signaling pathway by blocking downstream ERK signaling, which is regulated by MACC1 [49–51], This provides a rationale for combining MEK inhibitors with therapies targeting MACC1 to potentiate the therapeutic response in EC.

In addition to the MEK inhibitors, emerging small molecules targeting MACC1 directly are also gaining attention. Studies have demonstrated that MACC1 overexpression leads to increased tumor cell proliferation and metastasis [52–54]. There is mounting evidence that MACC1 contributes to the growth and spread of tumors [55,56]. The role of MACC1 in cancer progression and metastasis makes it a promising therapeutic target. As highlighted in our study, MACC1 is significantly overexpressed in EC cells, and its elevated levels correlate with poorer prognosis, suggesting its potential as a biomarker for prognosis and a target for treatment. While MACC1 has been well studied in several cancers, including colon, gastric, and lung cancers, its role in EC remains less explored. Our findings demonstrate that MACC1 knockdown effectively suppresses tumor migration, invasion, and proliferation, highlighting its functional importance in EC progression.

The integration of metabolic reprogramming as a therapeutic approach in EC is also becoming increasingly important. As discussed, metabolic shifts in EC cells, such as increased aerobic glycolysis and fatty acid synthesis, contribute to tumor progression [38,39]. Targeting these altered metabolic pathways in combination with MACC1 inhibition could further enhance treatment efficacy. In particular, the regulation of mitochondrial function and ROS production, both of which are influenced by MACC1, represents another potential avenue for therapeutic intervention [37].

A notable limitation of our study is the lack of external validation due to the current scarcity of publicly available endometrial cancer datasets with complete survival information. This highlights the need for more comprehensive public databases with detailed prognostic data for endometrial cancer patients, which would facilitate future validation studies and enhance the robustness of prognostic models. Additionally, although we conducted both in vivo and in vitro experiments, the precise molecular mechanism by which MACC1 promotes endometrial cancer progression remains to be fully elucidated. Further investigation is needed to comprehensively understand the role of MACC1 in EC pathogenesis, which could potentially improve diagnostic accuracy, treatment strategies, and prognostic evaluation.

## Supporting information

**S1 Fig. NMF clustering consensus maps.**
(TIFF)

**S2 Fig. Risk score distribution and survival status of EC patients in Train Set (A), Test Set (B) and Total Set (C).**
(TIFF)

**S3 Fig. The progression free survival (PFS) for patients in Train Set (A), Test Set (B) and Total Set (D).**
(TIFF)

**S4 Fig. The expression levels of ten MRGs between low-risk and high-risk groups.** (A) SAMM50, (B) NDUFAF6, (C) CMPK2, (D) MT-TP, (E) MAIP1, (F) TOMM40L, (G) MACC1, (H) DUSP18.
(TIFF)

**S5 Fig. The clinicopathological stratified analysis exploring the prognostic capacity of the MRG-related signature. (A-H)** The K-M survival curves of EC patients in the risk groups considering clinicopathology subgroups, including Age<60 **(A)**, Age>=60 **(B)**, Grade I-II **(C)**, Grade III **(D)**, Stage I-II **(E)**, Stage III-IV **(F)**, Cluster I **(G)**, and Cluster II **(H)**, LNM Negative **(I)**, and LNM Positive **(J)**.
(TIFF)

**S6 Fig. Evaluation of immune activity and immunotherapy for the two MRG-related risk groups. (A)** Analysis of immune activity between the two risk groups using CIBERSORT, EPIC, ESITMATE, MCPcounter, quanTIseq, TIMER and xCell. *$P<0.05$. **(B)** The subclass map showing the immunotherapeutic responses in different risk groups.
(TIFF)

**S7 Fig. The AUC of each model was calculated using 113 ML algorithm.**
(TIFF)

**S8 Fig. The frequency of MRGs observed in the 113 algorithm combinations(A).** The expressed of SAMM50**(B)**, MACC1**(C)**, DUSP18**(D)** and NDUFAF6**(E)** in EC.
(TIFF)

**S9 Fig. The original image of Fig 7B.**
(TIFF)

**S10 Fig. The original image of Fig 7D.**
(TIFF)

**S11 Fig. The original image of Fig 7G.**
(TIFF)

**S1 Table. Baseline characteristic of patients with endometrial cancer.**
(XLSX)

**S2 Table. The list of mitochondrial-related genes.**
(XLSX)

**S3 Table. The MRGs used for risk score.**
(XLSX)

**S4 Table. Primer sequence used in this study.**
(XLSX)

**S5 Table. Details of siRNA sequences.**
(XLSX)

## Acknowledgments

We greatly thank the data gained from The Cancer Genome Atlas (TCGA) (https://portal.gdc.cancer.gov/), the Genotype-Tissue Expression (GTEx) project (https://www.gtexportal.org/home/) and MitoCarta 3.0 database (https://www.broadinstitute.org/mitocarta/mitocarta30-inventory-mammalian-mitochondrial-proteins-and-pathways).

## Author contributions

**Conceptualization:** Qinying Liu, Yang Sun.

**Data curation:** Linying Liu.

**Validation:** Xuefen Lin, Yanhong Li.

**Visualization:** Xuefen Lin, Jianfeng Zheng, Jie Lin.

**Writing – original draft:** Xuefen Lin.

**Writing – review & editing:** Xuefen Lin, Jianfeng Zheng.

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
