## [Decision Letter · Decision Letter 0]

12 Nov 2024

Dear Dr. Sun,

Thank you for submitting your manuscript to PLOS ONE. After careful consideration, we feel that it has merit but does not fully meet PLOS ONE’s publication criteria as it currently stands. Therefore, we invite you to submit a revised version of the manuscript that addresses the points raised during the review process.

Please provide a point-by-point response to the reviewers' comments. Further, please edit illustrations to comply with journal guideline as highlighted by reviewer 1.

We look forward to receiving your revised manuscript.

Kind regards,

Peh Yean Cheah, Ph.D.

Academic Editor

PLOS ONE

Reviewers' comments:

Reviewer's Responses to Questions

**Comments to the Author**

1. Is the manuscript technically sound, and do the data support the conclusions?

Reviewer #1: Partly

Reviewer #2: Partly

2. Has the statistical analysis been performed appropriately and rigorously?

Reviewer #1: Yes

Reviewer #2: Yes

3. Have the authors made all data underlying the findings in their manuscript fully available?

Reviewer #1: Yes

Reviewer #2: Yes

4. Is the manuscript presented in an intelligible fashion and written in standard English?

Reviewer #1: Yes

Reviewer #2: Yes

Reviewer #1: The authors generated a prognostic signature for endometrial cancer patients and chose MACC1 as a key gene to further study in EC tumorigenesis. The study contains bioinformatic analysis and a figure on in vitro validation, and it would need to be further improved for its publication. There are some concerns that the authors should address:

Major concerns:

1) In abstract: “Key MRGs with important features were identified using 113 algorithms and validated through colony formation, EdU proliferation, wound healing, apoptosis, transwell assays, and animal studies”.

This line does not correlate with the figures presented here. There is only in vitro validation data for one MRG, MACC1, and there are no animal studies. In addition, there is a section for in vivo assays in materials and methods, and it is mentioned in several places. Still, in the last paragraph of the discussion the authors write “However, further in vivo experimental validation is necessary as this study has only been tested experimentally in vitro”. The authors need to explain and either present the in vivo data or adapt the text accordingly.

2) The authors need to specify number of patients and characteristics of the cohort they used, as well as number of patients for each group of the KM curves. In addition, they need to validate their results in another cohort (for example, others available at the R2 database).

3) Lines 370-375. Please rephrase, it is not clear what the authors want to communicate here nor which the results they obtained are.

4) In general, the study needs to be explained better and the results more thoroughly described. It would also benefit from simplifying the figures, moving some to supplementary information, and homogenizing sizes and font sizes, and including a graphical abstract as summary.

For instance, Figure 8A is hard to understand and follow, and in the text it is not described what the results actually are, only that “there are differences” (lines 413-417). The authors should describe these results and try to communicate the main points to the reader.

5) Lines 432-436. The authors describe “According to our findings, a positive correlation was observed between the levels of MACC1 and CMPK2 and several drugs, including vinorelbine, cytarabine, docetaxel, tamoxifen, sorafenib, and crizotinib (Figure 9I). Conversely, an inverse relationship was identified between the levels of SAMM50 and NDUFAF6 and certain medications such as Paclitaxel and Sorafenib (Figure 9I)”.

The correlation is not between genes and drugs, but with drug susceptibility. Please correct the language.

6) Lines 438-468. The title does not fit with the results since the panel “N” in Figure 10 does not exist. The authors should explain if this is a mistake or if they decided to remove the in vivo data and forgot to adapt the text.

7) Fig. 10F-G. The authors need to include additional EC cell lines, to have a panel that clearly shows that MACC1 is elevated in this type of cancer.

Fig. 10 H-M. It would be convenient to show these experiments for si1, even if the knockdown efficiency is not as good as for si2, but it will prove the correlation between the levels of MACC1 and the effects on tumorigenesis.

Figure 10I. This assay was performed at 14 days. Usually, the siRNAs last for a week or less. The authors should show a time-curse WB demonstrating that the knockdown of MACC1 is maintained throughout the experiment.

8) In the discussion, please include information on the current literature regarding metabolism in endometrial cancer, and the importance of your study in relation to it. In addition, please include a discussion on the possible therapeutic applicability of your study, and how it would be possible to target MACC1 (for instance, with MEK inhibitors).

Minor concerns:

- Line 90: Spell out “EEC”.

- Line 186: Correct to “Cell culture”

- Figure 10J: Correct to “Hoechst” and include scale bar.

Reviewer #2: In this manuscript, the authors create a prognostic model for endometrial cancer using mitochondria-related genes (MRGs) to predict patient outcomes, drug sensitivity, and immunotherapy response. They identify of the gene MACC1 as a marker for endometrial cancer progression, demonstrating that MACC1 downregulation reduces proliferation, migration and invasion, making it a potential therapeutic target. I have the following criticisms that the authors might address:

Major comments:

- Introduction, lines 115 – 117, the authors write: “however, it remains unclear how individuals with endometrial cancer would respond to genes associated with these organelles”. What do the authors mean by responding to genes? Respond to mutations or if these genes are targeted?

- It is unclear if in Figure 1A the MRGs in EC were compared to healthy tissue. Otherwise, how can you discern if a gene is upregulated or downregulated?

- It would be of great help for the reader if the authors spelled out the MRGs selected for the risk score. In addition, for MACC1, a description of its function would be beneficial also in the results part to understand the following experiments in vitro.

- In Figure 10B, the authors analyze in silico the expression of the different selected markers (SAMM50, NDUFAF6, MACC1, and DUSP18) in normal vs tumor tissues. In accordance with their previous results, these markers should be higher expressed in high-risk samples. Is the expression of these genes elevated in higher tumor stages compared to low risk classified tumors? This could be assessed by dividing the tumors in low and high risk and comparing gene expression.

- Figure 10N, regarding the impact on in vivo tumor growth of MACC1 downregulation, is not included in the submitted version.

- In the discussion, the manuscript would benefit from a correlation between the results observed in the in vitro experiments downregulating MACC1 and what it is known about its function regulating migration, epithelial to mesenchymal transition and cytoskeleton dynamics.

- Even though the authors describe in vivo work in the last part of the results, the methods section of that experiment is described in detailed, and mention Figure 10N which is not included, in the discussion it is written that this study has only been tested experimentally in vitro.

Minor comments:

- Introduction, line 99 – 100, the authors write: “Mitochondria take up substrates from the cytoplasm and use them for important life activities(6).” In this context, it would be better to change “life activities” for “cellular processes”

- Materials and methods, line 186: correct to cell culture.

**Do you want your identity to be public for this peer review?** For information about this choice, including consent withdrawal, please see our Privacy Policy

Reviewer #1: No

Reviewer #2: No

---

## [Author Response · Author response to Decision Letter 1]

4 Feb 2025

We gratefully appreciate the editors and all the reviewers for their time spend making their constructive remarks to our manuscript entitled “Mitochondria-related genes as prognostic signature of endometrial cancer and the effect of MACC1 on tumor cells”. Those comments are all valuable and very helpful for revising and improving our manuscript, as well as the important guiding significance to our study. The comments have been carefully studied and we have made necessary corrections in the hope of gaining approval. We have submitted our manuscript with tracked changes to highlight the revisions.

The Point-by-point response to reviewers are as follows:

Reviewer 1

Comment

The authors generated a prognostic signature for endometrial cancer patients and chose MACC1 as a key gene to further study in EC tumorigenesis. The study contains bioinformatic analysis and a figure on in vitro validation, and it would need to be further improved for its publication. There are some concerns that the authors should address:

Major concerns:

1) In abstract: “Key MRGs with important features were identified using 113 algorithms and validated through colony formation, EdU proliferation, wound healing, apoptosis, transwell assays, and animal studies”.

This line does not correlate with the figures presented here. There is only in vitro validation data for one MRG, MACC1, and there are no animal studies. In addition, there is a section for in vivo assays in materials and methods, and it is mentioned in several places. Still, in the last paragraph of the discussion the authors write “However, further in vivo experimental validation is necessary as this study has only been tested experimentally in vitro”. The authors need to explain and either present the in vivo data or adapt the text accordingly.

2) The authors need to specify number of patients and characteristics of the cohort they used, as well as number of patients for each group of the KM curves. In addition, they need to validate their results in another cohort (for example, others available at the R2 database).

3) Lines 370-375. Please rephrase, it is not clear what the authors want to communicate here nor which the results they obtained are.

4) In general, the study needs to be explained better and the results more thoroughly described. It would also benefit from simplifying the figures, moving some to supplementary information, and homogenizing sizes and font sizes, and including a graphical abstract as summary.

For instance, Figure 8A is hard to understand and follow, and in the text it is not described what the results actually are, only that “there are differences” (lines 413-417). The authors should describe these results and try to communicate the main points to the reader.

5) Lines 432-436. The authors describe “According to our findings, a positive correlation was observed between the levels of MACC1 and CMPK2 and several drugs, including vinorelbine, cytarabine, docetaxel, tamoxifen, sorafenib, and crizotinib (Figure 9I). Conversely, an inverse relationship was identified between the levels of SAMM50 and NDUFAF6 and certain medications such as Paclitaxel and Sorafenib (Figure 9I)”.

The correlation is not between genes and drugs, but with drug susceptibility. Please correct the language.

6) Lines 438-468. The title does not fit with the results since the panel “N” in Figure 10 does not exist. The authors should explain if this is a mistake or if they decided to remove the in vivo data and forgot to adapt the text.

7) Fig. 10F-G. The authors need to include additional EC cell lines, to have a panel that clearly shows that MACC1 is elevated in this type of cancer.

Fig. 10 H-M. It would be convenient to show these experiments for si1, even if the knockdown efficiency is not as good as for si2, but it will prove the correlation between the levels of MACC1 and the effects on tumorigenesis.

Figure 10I. This assay was performed at 14 days. Usually, the siRNAs last for a week or less. The authors should show a time-curse WB demonstrating that the knockdown of MACC1 is maintained throughout the experiment.

8) In the discussion, please include information on the current literature regarding metabolism in endometrial cancer, and the importance of your study in relation to it. In addition, please include a discussion on the possible therapeutic applicability of your study, and how it would be possible to target MACC1 (for instance, with MEK inhibitors).

Minor concerns:

Line 90: Spell out “EEC”.

Line 186: Correct to “Cell culture”

Figure 10J: Correct to “Hoechst” and include scale bar.

Response

Dear Reviewer,

We gratefully appreciate you for your time spend making your constructive remarks to our manuscript. Your comments are all valuable and very helpful for revising and improving our manuscript, as well as the important guiding significance to our study. The comments have been carefully studied and we have made necessary corrections in the hope of gaining approval. We have submitted our manuscript with tracked changes to highlight the revisions.

Comment 1

In abstract: “Key MRGs with important features were identified using 113 algorithms and validated through colony formation, EdU proliferation, wound healing, apoptosis, transwell assays, and animal studies”.

This line does not correlate with the figures presented here. There is only in vitro validation data for one MRG, MACC1, and there are no animal studies. In addition, there is a section for in vivo assays in materials and methods, and it is mentioned in several places. Still, in the last paragraph of the discussion the authors write “However, further in vivo experimental validation is necessary as this study has only been tested experimentally in vitro”. The authors need to explain and either present the in vivo data or adapt the text accordingly.

Response 1

We sincerely appreciate your constructive comments. We would like to address these concerns as follows:

1. Regarding the MRG validation:

- We have revised the phrasing in the abstract to clarify that MACC1 was selected as the key MRG for experimental validation based on our comprehensive bioinformatic analysis.

- The rationale for selecting MACC1 is thoroughly explained in the Results section.

Line 483-513

2. Regarding the animal experimental data:

- Due to our oversight during submission, we inadvertently omitted the figure containing animal experimental data.

- We have now included these data as Fig 8 in the revised manuscript.

3. We have carefully reviewed and revised the entire manuscript to ensure:

- Consistent description of the experimental validation throughout the text.

- Accurate representation of both in vitro and in vivo findings.

- Removal of the incorrect statement in the discussion about "only in vitro validation".

Line 40-43:

Systematic experimental validation, including both in vitro and in vivo approaches, demonstrated that MACC1 downregulation significantly suppressed EC progression, highlighting its potential as a therapeutic target.

Line 632-637:

Additionally, although we conducted both in vivo and in vitro experiments, the precise molecular mechanism by which MACC1 promotes endometrial cancer progression remains to be fully elucidated. Further investigation is needed to comprehensively understand the role of MACC1 in EC pathogenesis, which could potentially improve diagnostic accuracy, treatment strategies, and prognostic evaluation.

Comment 2

2) The authors need to specify number of patients and characteristics of the cohort they used, as well as number of patients for each group of the KM curves. In addition, they need to validate their results in another cohort (for example, others available at the R2 database).

Response 2

We appreciate your constructive comments. We acknowledge the oversight in our original manuscript regarding patient information and have made the following clarifications:

1. Patient Information and Cohort Characteristics:

- The total number of participants in our study was 545 primary EC patients and 113 healthy controls.

- The detailed patient characteristics (including age, FIGO stage, grade, etc.) have been added to S1 Table in the revised manuscript.

- For the KM curves analysis, patient numbers are clearly shown below each curve.

2. Regarding External Validation:

We made extensive efforts to validate our findings using external datasets:

a) Public Database Search:

- GEO database (https://www.ncbi.nlm.nih.gov/geo/)

- R2 Genomics Analysis Platform (https://hgserver1.amc.nl/cgi-bin/r2/main.cgi)

- TCGA and GTEx databases (which we have already included in our analysis)

b) Commercial Microarray Resources:

- Outdo Biotech Ltd (https://outdoivd.com/biology/cdna.html)

- DAIXUE Bio Ltd (http://www.biotechdx.com/plus/list.php?tid=16)

- Shanghai Biochip Ltd (https://www.shbiochip.com/wzsy)

- Beijing Bionacil Biotechnology Ltd (http://bionaxin.com/sy)

Unfortunately, none of these sources contained endometrial cancer datasets with the necessary prognostic information for model validation. This limitation is primarily due to the current scarcity of publicly available endometrial cancer datasets with complete survival data. We have acknowledged this limitation in the Discussion section of our manuscript.

Comment 3

3) Lines 370-375. Please rephrase, it is not clear what the authors want to communicate here nor which the results they obtained are.

Response 3

We appreciate your constructive comments. To address this concern, we have thoroughly revised this section to enhance clarity. The core objective of this analysis was to evaluate the prognostic value of our MRG-related signature across different clinicopathological subgroups. We have modified the text as follows:

Line 378-385:

To comprehensively validate the prognostic robustness of our MRG-related signature, we conducted stratified survival analyses across key clinicopathological features, including age (<60 vs. ≥60 years, S5 Fig A-B), tumor grade (G1-2 vs. G3, S5 Fig C-D), FIGO stage (I-II vs. III-IV, S5 Fig E-F), MRG-related clusters (S5 Fig G-H), and lymph node status (positive vs. negative, S5 Fig I-J). Notably, the signature maintained its prognostic significance within each subgroup, consistently demonstrating significantly poorer overall survival in high-risk patients compared to their low-risk counterparts (all P < 0.05).

Comment 4

4) In general, the study needs to be explained better and the results more thoroughly described. It would also benefit from simplifying the figures, moving some to supplementary information, and homogenizing sizes and font sizes, and including a graphical abstract as summary.

For instance, Figure 8A is hard to understand and follow, and in the text it is not described what the results actually are, only that “there are differences” (lines 413-417). The authors should describe these results and try to communicate the main points to the reader.

Response 4

We sincerely appreciate your constructive suggestions for improving the manuscript's clarity and presentation. We have implemented comprehensive changes as follows:

1. Figure Organization and Presentation:

- Reorganized figures by moving detailed technical analyses to supplementary materials.

- Standardized all figures following PLOS ONE guidelines, including consistent font sizes and formatting.

- Created a graphical abstract to provide a clear visual summary of our study design and key findings.

2. Enhanced Results Description:

Lines 430-453 have been revised to provide a more detailed explanation:

"To systematically characterize the immune landscape associated with our risk signature, we employed seven complementary computational deconvolution methods (CIBERSORT, EPIC, ESTIMATE, MCPcounter, quanTIseq, TIMER, and xCell) to profile the tumor immune microenvironment (S6 Fig A). This multi-algorithmic approach enabled robust and comprehensive evaluation of immune cell infiltration patterns. Our analysis revealed striking differences in immune cell composition between high- and low-risk groups. Notably, we observed significant enrichment of multiple immune cell populations in the high-risk group, particularly in adaptive immune components. Specifically, both CD4+ and CD8+ T cell populations showed consistent elevation across multiple algorithms (MCPcounter_T_cells, MCPcounter_CD8_T_cells), suggesting enhanced T cell infiltration in high-risk tumors. Intriguingly, we observed substantial differences in stromal components, particularly in endothelial cells and cancer-associated fibroblasts (EPIC_CAFs, EPIC_Endothelial, MCPcounter_Endothelial_cells, xCellmv_Endothelial_cells). This finding suggests that the risk signature not only reflects immune cell dynamics but also encompasses broader changes in the tumor microenvironment. The ESTIMATE scores further corroborated these findings, indicating significant differences in both immune and stromal components between risk groups. Of particular interest was the enhanced NK cell signature (MCPcounter_NK_cells) in the high-risk group, suggesting increased natural cytotoxicity potential. This observation, combined with the elevated T cell signatures, indicates a more active but potentially dysregulated immune response in high-risk tumors. Notably, these findings suggest that high-risk patients might be particularly responsive to immunotherapy approaches, specifically PD1-R therapy, as supported by our additional analysis (S6 Fig B)."

3. Figure Simplification:

- Former Figure 8A (now S6 Fig A) has been redesigned for better clarity

- Added clear annotations and legends

- Included statistical significance indicators

- Provided a detailed figure legend explaining the analysis methods and key findings.

These revisions aim to enhance the manuscript's readability while ensuring that our scientific findings are presented in a clear and accessible manner.

Comment 5

5) Lines 432-436. The authors describe “According to our findings, a positive correlation was observed between the levels of MACC1 and CMPK2 and several drugs, including vinorelbine, cytarabine, docetaxel, tamoxifen, sorafenib, and crizotinib (Figure 9I). Conversely, an inverse relationship was identified between the levels of SAMM50 and NDUFAF6 and certain medications such as Paclitaxel and Sorafenib (Figure 9I)”.

The correlation is not between genes and drugs, but with drug susceptibility. Please correct the language.

Response 5

Thank you for this important observation regarding the precise terminology in describing drug-gene relationships. We fully agree that accuracy in describing pharmacogenomic relationships is crucial, particularly in the context of drug susceptibility and treatment response. We have revised the text to more accurately reflect the relationship between gene expression levels and drug susceptibility:

Line 465-474:

According to our findings, the genes in the model were associated with drug susceptibility. A positive correlation was observed between the levels of MACC1 and CMPK2 and the drug resistance scores calculated by oncoPredict for several medications, including vinorelbine, cytarabine, docetaxel, tamoxifen, sorafenib, and crizotinib (Figure 6I). This indicates that higher expression levels of MACC1 and CMPK2 may be associated with decreased drug sensitivity to these drugs. Conversely, an inverse relationship was identified between the levels of SAMM50 and NDUFAF6 and the drug resistance scores for certain medications, such as Paclitaxel and Sorafenib (Figure 6I). This suggests that higher expression levels of SAMM50 and NDUFAF6 may be linked to increased drug sensitivity to these drugs.

Comment 6

6) Lines 438-468. The title does not fit with the results since the panel “N” in Figure 10 does not exist. The authors should explain if this is a mistake or if they decided to remove the in vivo data and forgot to adapt the text.

Response 6

Thank you for your careful observation regarding this discrepancy. We sincerely apologize for this oversight in our submission. You are correct that there was a mismatch between the manuscript text and Figure 10, where panel "N" was referenced but not included. This occurred during th

---

## [Decision Letter · Decision Letter 1]

1 Apr 2025

Mitochondria-related genes as prognostic signature of endometrial cancer and the effect of MACC1 on tumor cells

PONE-D-24-32704R1

Dear Dr. Sun,

We’re pleased to inform you that your manuscript has been judged scientifically suitable for publication and will be formally accepted for publication once it meets all outstanding technical requirements.

Kind regards,

Yuanliang Yan

Academic Editor

PLOS ONE

Additional Editor Comments (optional):

Reviewers' comments:

Reviewer's Responses to Questions

**Comments to the Author**

Reviewer #1: All comments have been addressed

Reviewer #2: All comments have been addressed

2. Is the manuscript technically sound, and do the data support the conclusions?

Reviewer #1: Yes

Reviewer #2: Yes

3. Has the statistical analysis been performed appropriately and rigorously?

Reviewer #1: Yes

Reviewer #2: Yes

4. Have the authors made all data underlying the findings in their manuscript fully available?

Reviewer #1: Yes

Reviewer #2: Yes

5. Is the manuscript presented in an intelligible fashion and written in standard English?

Reviewer #1: Yes

Reviewer #2: Yes

Reviewer #1: The authors have addressed all concerns and now the conclusions are better supported by the data, as well as the experiments made more thoroughly.

Please correct in Figure 8D "tumer" to "tumor" in the axis.

Reviewer #2: The authors have addressed all the critiques carefully and thoroughly, and the current manuscript has more solid data on describing a prognostic model for endometrial cancer using mitochondrial-related genes, with special focus on the MACC1 gene. Just a small correction should be done before publication in PLOS One: please correct the word "tumer" to "tumor" in Figure 8D.

**Do you want your identity to be public for this peer review?** For information about this choice, including consent withdrawal, please see our Privacy Policy

Reviewer #1: No

Reviewer #2: No

---

## [Editor Report · Acceptance letter]

PONE-D-24-32704R1

PLOS ONE

Dear Dr. Sun,

I'm pleased to inform you that your manuscript has been deemed suitable for publication in PLOS ONE. Congratulations! Your manuscript is now being handed over to our production team.

Kind regards,

on behalf of

Dr. PLOS Manuscript Reassignment

Staff Editor

PLOS ONE